# Deformation Prediction and Experimental Study of 316L Stainless Steel Thin-Walled Parts Processed by Additive-Subtractive Hybrid Manufacturing

**DOI:** 10.3390/ma14195582

**Published:** 2021-09-26

**Authors:** Xuefeng Wu, Wenbo Zhu, Yu He

**Affiliations:** 1School of Mechanical and Power Engineering, Harbin University of Science and Technology, Harbin 150080, China; zhu13997616260@163.com; 2School of Mechatronics Engineering, Harbin Institute of Technology, Harbin 150001, China; hitheyu2016@126.com

**Keywords:** 316L stainless steel, additive-subtractive hybrid manufacturing, laser melting deposition, thin-walled parts, stress and deformation

## Abstract

The hybrid process integrates two or more different processes, such as additive and subtractive manufacturing, which have gained appreciable consideration in recent years. The deformation of hybrid manufacturing is an essential factor affecting machining quality. The purpose of this paper is to study the effect of milling on stress release and surface deformation of additive manufacturing (AM) specimens in the process of additive and subtractive hybrid manufacturing (ASHM) of 316L stainless steel thin-walled parts, so as to effectively improve the forming quality of thin-walled parts manufactured by the combined processing of ASHM. To this end, a series of experiments were carried out to study the relationship between stress distribution and thermal stress deformation of 316L stainless steel thin-walled parts prepared by LMD, and the changes of stress and deformation of these thin-walled parts after subsequent milling. An infrared camera and laser distance sensor were used to record the temperature field data and deformation data to analyze the influence factors of temperature and stress on the machining results. Then, the finite element software was used to simulate the stress and deformation of the thin-walled parts in the additive manufacturing process and the subsequent milling process. Meanwhile, the model was verified through the experiments. In addition, the relationship between the milling force and the milling parameters of the AM parts was studied by orthogonal test and regression analysis.

## 1. Introduction

Laser hybrid additive-subtractive manufacturing is a new type of rapid direct forming technology. The technology relies on Laser Melting Deposition (LMD), which combines Additive Manufacturing (AM) and traditional Subtractive Manufacturing (SM) to produce parts with complex geometry, good surface finish, and dimensional accuracy. Among the different metal additive manufacturing technologies available, the industry has predominantly opted for Powder Bed Fusion (PBF) and Directed Energy Deposition (DED) processes. Almost any weldable metal can be processed with any of these two techniques. Nevertheless, most hybrid systems integrate LMD, a DED technology that is faster than Selective Laser Melting (SLM), a PBF process, and does not need any process chamber or supporting structures [1]. However, the residual stress generated by these two processes is a challenge for thin-walled parts. On the one hand, in the laser additive manufacturing process, as the high energy laser beam moving back and forth and molten metal powder stacking layer upon layer on the substrate, additively manufactured parts repeatedly experience a sudden warming and cooling process. The temperature gradient change for a long time will produce enormous internal stress, which will cause cracks and warpage deformation of thin-walled parts.

On the other hand, although the subsequent subtractive manufacturing process will release part of the additive thermal stress to a certain extent, the contact between the workpiece and the tool will produce plastic deformation and cutting heat during the cutting process. It is inevitable to introduce machining residual stress on the workpiece surface. Moreover, under the influence of cutting force, cutting heat, cutting vibration, and other factors, thin-walled parts are elementary to deform and reduce dimensional accuracy [2].

At present, the processing parameters of LMD and the rapid, alternating temperature cycle effect on the microstructure and macroscopic morphology of AM parts [3], along with controlling the thermal stress in AM and residual stress in SM, still restricts the development of ASHM. Researchers have made many efforts for this purpose. Qian Bai et al. [4] studied the milling force, surface roughness, and residual stress of 6511 stainless steel samples by SLM combined with end milling. The results show that milling parameters can control the state and distribution of surface residual stress. Peirong Zhang [5] systematically studied the effects of the continuous turning and polishing process on surface morphology, residual stress, microhardness, porosity, and bonding strength of laser cladding layer of Cr-Ni based stainless steel. The results show that the continuous polishing process can optimize the surface morphology and improve the microhardness. The compressive residual stress increases the porosity and bonding strength of the cladding layer. Mengqiu Gao [6] studied the effect of substrate temperature state on surface quality of a 316L stainless steel workpiece in the process of ASHM. They milled AM parts that cooled for different periods and measured the workpiece’s surface roughness, residual stress, and microhardness.

In addition to a single-layer cladding workpiece, block, and standard tensile specimen, most of the formed parts in the ASHM are thin-walled parts, such as the single-channel thin wall, thin-walled ring, thin-walled frame, etc. It has been found that the residual stress on the outer surface of the specimen manufactured by ASHM is less than that of the LMD because the milling releases part of the residual stress, and the macroscopic distribution of the residual stress on the specimen surface is not uniform. Yuying Yang et al. [7] conducted an experimental study on the distribution of residual stress on the surface of a 316L stainless steel thin-walled ring manufactured by ASHM by using the X-ray diffraction method. The results show that the top and bottom of the sample surface are tensile stress, while the middle part is compressive stress, and the stress value gradually increases with the increase of the height of the deposition layer. Bo Xin et al. [8] choose different *Z*-axis lifting capacities to form thin-walled stainless steel parts, which are processed by powder-feeding laser deposition forming technology. Thin-walled parts were detected in different parts of the microstructure and mechanical properties, including grain size, depth of the remelting, and tensile properties of anisotropic. They also explore the single-channel cladding layer and the grain morphology and growth direction of the thin-walled parts as a whole.

In addition to the experimental analysis of ASHM, a large number of numerical simulations have been carried out to investigate the effects of AM process parameters and component sizes on the distribution and control of thermal stress and residual stress in AM parts.

ANSYS is used to establish the monolayer cladding model of 316L stainless steel and to simulate and analyze its temperature and stress fields. Results show that the residual stress is mainly reflected in the vertical direction of the cladding, which is primarily manifested as tensile stress. With the increase of the depth of the cladding layer, the tensile stress first increases and then decreases, and the maximum tensile stress is located at 0.2 mm away from the substrate. Vasinonta et al. [9] established a two-dimensional thermodynamic model of the thin-walled structure of 304 stainless steel by LMD, studied the effects of laser power, scanning speed, and other process parameters on the size of molten pool and the residual stress distribution of the deposited wall. Somashekara et al. [10] used finite element numerical simulation to predict the influence of additive manufacturing process parameters on thermal stress and residual stress. The calculation results showed that the thermal stress at the top layer of AM was less than that at the bottom layer. However, the distribution law of residual stress was opposite, and the residual stress at the bottom layer was relatively small. Steen [11] established a 3D thermal finite element model using the “birth and death elements” technology. This model took heat conduction as the only heat transfer mode. The thermal behavior of the thin-walled deposition process is analyzed. They set the element node in the molten pool as the melting or overheating temperature to simulate the heat input. It was determined the thermal gradient position relative to the geometric shape of the parts. It plays a guiding role in reducing deformation and thermal stress in parts manufacturing.

Although simulation can be used to predict residual stresses and deformations, thus reducing or compensating for them, it is not clear how subsequent processing of immense stress AM parts affect the deformation of the parts [12].

In the process of laser metal deposition (LMD), thermal stress has a great influence on the forming quality of thin-walled parts. At present, most of the studies on the deformation of thin-walled parts made by LMD are one-sided, especially the systematic research on the thermal deformation of thin-walled parts in AM and the further influence of stress release and milling force on the deformation in later subtractive manufacturing is lacking. Therefore, we conducted a careful study on this issue. In this paper, 316L stainless steel powder was used as the material to experiment with laser cladding thin-walled parts with ASHM. The thermal stress deformation of the thin-walled parts manufactured by AM and the deformation of these parts after milling were measured, respectively. Then, based on the finite element software ANSYS, a 3D sequentially coupled thermodynamic model was established to simulate the transient temperature field and stress distribution during the hybrid manufacturing process of 316L stainless steel thin-walled parts. This model was also used to predict the final deformation of the thin-walled parts. The comparison with the experimental results proves the accuracy of the finite element model, which provides a reference for effectively controlling the deformation of thin-walled parts after milling in AM. The research content of this paper is shown in Figure 1.

## 2. Experiment on the Fabrication of 316L Stainless Steel Thin-Walled Parts by LMD

### 2.1. Experiment Setup

The experimental system mainly includes a three-axis machine tool (Beijing Beiyi Machine Tool Co., Ltd., Beijing, China), YLS-2000 fiber laser (IPG, Oxford, MA, America), double cylinder powder feeder (Raycham, Nanjing, China), and YC52 coaxial powder feeding laser cladding head (PRECITEC, Gaggenau, Germany). In addition, there is auxiliary equipment such as a water cooling machine (IPG, Oxford, MA, America), argon gas supply system, etc. The temperature field of AM was recorded by an Optris automatic hot spot tracking infrared camera (Optris, Berlin, Germany) with an accuracy of ±2 °C. HG-C1030 laser distance sensor produced by Panasonic (Beijing, China) is used to detect the surface deformation of the workpiece. The measuring range is 30 ± 5 mm, and the accuracy is 10 μm. It can detect the deformation of the AM parts online in time in the ASHM. The composition of the whole additive and subtractive hybrid manufacturing system is shown in Figure 2. As shown in Figure 2, the main body of the entire ASHM system is a three-axis milling machine. The laser cladding head for AM is installed on the *Z*-axis side of the milling machine. The movement of the shaft moves. The positioning accuracy of the machine tool is ±0.015/300 mm, and the repeat positioning accuracy is ±0.005 mm.

The powder used in the LMD is spherical 316L stainless steel metal powder. The main chemical composition is C, Si, Mn, Cr, Ni, Mo, Fe, its granularity is 75~150 μm, has good powder fluidity and technology. The chemical composition of 316L stainless steel powder is shown in Table 1. The material of the substrate is a 316L rolled stainless steel plate with a size of 250 mm × 150 mm × 15 mm. Before the experiment, it is necessary to dry the metal powder in a vacuum drying oven at 100 °C for 4 h to ensure that the powder has good fluidity. Alcohol should be used to wipe the surface of the substrate before the experiment to remove grease and dust.

### 2.2. Experiment on Single-Pass Cladding Layer Forming

The essence of laser cladding forming thin-walled parts is the accumulation of multi-layer single-pass cladding layers. Therefore, if the forming quality of the single-pass cladding layer is lacking, the moral defects of single-layer cladding will appear and accumulate many times in the accumulation process, which will seriously affect the forming accuracy and internal quality of thin-walled parts. Therefore, it is essential to carry out a single-pass layer forming experiment before AM of thin-walled parts.

Parameters including laser parameters, powder feeder parameters, gas supply system parameters, laser deposition head parameters, and the machine tool parameters need to be adjusted in the laser cladding forming single-pass layer experiments. The optimized laser cladding parameters have been found out through tests prepared for the following laser cladding forming thin-walled parts. The corresponding adjustable parameters are shown in Table 2. This paper mainly considers the influence of laser power, *Z*-axis lifting amount, and scanning interval time between layers on the forming morphology of the cladding layer.

According to the study of Yu He et al. [13], laser power significantly impacts the size of the single-pass cladding layer. While for the forming quality of thin-walled parts, the change of scanning interval time between layers and *Z*-axis lifting has a more significant impact. The experiment results show that in the process of single-pass, single-layer cladding forming, under the same protective atmosphere, the single-layer cladding formed by 600 W laser power has better surface quality, less oxidation, and is brighter overall, as shown in Figure 3a. In the process of single-pass, multi-layer (thin-walled parts) forming, too long scanning interval time between layers will lead to a collapse at both ends of the thin wall (as seen in Figure 3b-①). If the *Z*-axis lifting is too large, the overall height of the thin-walled parts, especially the middle part, will decrease (see Figure 3b-③).

Therefore, after many tests and comparisons, the final selected process parameters related to laser additive manufacturing of 316L stainless steel thin-walled parts and the size of the thin-walled parts are shown in Table 3.

The laser power is fixed at 520 W due to the attenuation in the process of laser generation and transmission. The *Z*-axis lifting is the height that the cladding head needs to be lifted along the *Z*-axis after the previous cladding layer is formed. The *Z*-axis lifting should be kept within a reasonable range, no higher than the height of the first cladding layer. Otherwise, the forming size will not reach the expected target; simultaneously, the *z*-axis lifting should not be too small. Otherwise, it will cause equipment safety hazards [13]. The “zigzag” scanning strategy was used in the forming process. This method can avoid the uneven heating at both ends of thin-walled parts due to the single scanning direction, which leads to the forming quality reduction of thin-walled parts.

The laser power is fixed at 520 W due to the attenuation in the process of laser generation and transmission. The *Z*-axis lifting is the height that the cladding head needs to be lifted along the *Z*-axis after the previous cladding layer is formed. The *Z*-axis lifting should be kept within a reasonable range, not higher than the height of the first cladding layer. Otherwise, the forming size will not reach the expected target; simultaneously, the *z*-axis lifting should not be too small. Otherwise, it will cause equipment safety hazards [13]. The “zigzag” scanning strategy was used in the forming process. This method can avoid the uneven heating at both ends of thin-walled parts due to the single scanning direction, which leads to the forming quality reduction of thin-walled parts.

The infrared camera recorded the temperature field during the whole AM process of thin-walled parts. The emissivity of the infrared camera was calibrated by thermocouple before recording to ensure temperature measurement accuracy. After the AM process, the supporting software extracted the temperature change history of a point and compared it with the subsequent finite element simulation results.

At the end of the AM process, after the thin-walled parts are cooled to the ambient temperature, the residual powder accumulated around the bottom was cleaned with a brush. Then, the laser distance sensor is used to scan the surface of thin-walled parts. After processing by the analysis software, the deformation data of the surface of the thin-walled parts relative to the datum surface can be obtained.

### 2.3. Experiments Results and Analysis

#### 2.3.1. Analysis of Deformation of Thin-Walled Parts Manufactured by AM

The laser cladding forming process of 316L stainless steel thin-walled parts and the morphology of the final formed parts are shown in Figure 4. It can be seen from Figure 4b that the surface of 316L stainless steel thin-walled parts formed by laser cladding has noticeable stratification marks, and the overall appearance is wide at the top and narrow at the bottom. As shown from Figure 4b,d, the thin-walled parts are uneven along the cladding direction. Excessive accumulation occurs at both ends of the thin-walled parts, and the forming quality is poor compared with the middle position. Both ends of the top of the thin-walled parts are convexly deformed along the cladding direction, and the vertical cladding direction and the top ends of the thin-walled parts are recessed inward at a distance of about 2~12 mm from the substrate.

After the thin-walled parts had cooled to room temperature, the laser distance sensor was used to scan the surface of the thin-walled parts at different positions, as shown in Figure 5. The starting point of the cladding layer there was taken as the scanning reference point (marked with the red dot in Figure 5, and the deformation data of the bottom, middle, and top of the thin-walled parts were recorded, respectively). The line chart of the scanning data is shown in Figure 6a, and the box-plot of the deformation statistics of each part is shown in Figure 6b.

As shown from Figure 6b, except for a few outliers beyond the upper and lower edges of the box diagram, the thin-walled part has convex deformation relative to the datum level in these three positions. The deformation at the top position is more severe than the others, ranging from 0.130 mm to 1.412 mm. The deformation at the bottom is gentle, and the deformation range is the smallest. Among the three scanning sites, the maximum deformation value appeared in the middle, and the maximum deformation value was 1.554 mm.

For the offset from the reference measured in Figure 6a, in order to determine that (1) the location is a statistically relevant factor and (2) the differences between each pair (bottom vs. top/top vs. middle, etc.) are also statistically relevant, we conducted one-way ANOVA (Analysis of Variance) analysis and post hoc multiple comparisons (Tamhane’s T2) on the deformation of the top, middle and bottom of thin-walled parts. SPSS was used for data processing, and the results are as follows (Figure 7):

#### 2.3.2. Analysis of Temperature Change of Thin-Walled Parts Manufactured by AM

The infrared thermal imager (Optris, Germany) can observe the temperature distribution of the laser heat source in the thin-walled part during the AM process. Figure 8 shows the temperature distribution on the thin-walled part at a certain point in the AM process recorded by the infrared thermal imager and the change of temperature over time at the position shown by the cross cursor in Figure 8 during the entire AM process. Experimental data with high temperatures cannot be obtained due to the limitation of the infrared camera’s measuring range (150–900 °C). Therefore, the extrapolation method is used to fit the temperature data above 900 °C. The red dotted line in the curve is the fitting point.

The results show that the thin-walled parts experienced several heating and cooling processes during the AM process, and the highest temperature was 1459.4 °C. The temperature peak increases gradually before the arrival of the laser heat source. It decreases gradually after the arrival of the laser heat source, and the temperature fluctuation tends to be stable gradually. The cooling process gradually decreases with the temperature difference from the ambient temperature. The cooling curve of the thin-walled parts is approximately one of the inverse proportional function curves. The thin-walled parts experienced a rapid cooling stage in a short time after the end of the AM and a slow cooling stage, after which the cooling rate gradually decreased.

### 2.4. Summary

Large deformation occurred in the process of LMD of thin-walled parts, and powder adhesion also appeared in the upper middle part of the thin-walled parts (see Figure 5). The latter can be improved by optimizing the AM process parameters. The former is due to the long time temperature gradient change in the AM process, resulting in great thermal stress inside the thin-walled parts, which makes the thin-walled parts have convex or concave deformation compared with the design size. Therefore, single AM restricts the dimensional accuracy of thin-walled parts and affects their forming quality. Therefore, it is necessary to introduce milling to improve the forming quality of the thin-walled parts made by AM. Before that, the mechanism of thermal stress distribution and thermal deformation of thin-walled parts needs to be studied.

## 3. Finite Element Simulation of 316L Stainless Steel Thin-Walled Parts Manufactured by LMD

In this section, based on the finite element software ANSYS, a three-dimensional sequential coupling thermodynamic model was established to simulate the transient temperature field, stress distribution, and final deformation of 316L stainless steel thin-walled parts during the laser cladding manufacturing process. The finite element numerical simulation mainly includes two main steps: (1) conduct transient thermal analysis to generate the temperature history file of the whole laser additive remanufacturing process; and (2) perform mechanical analysis to calculate the residual stress and deformation in the laser additive remanufacturing process. The loading load of the step is the temperature history record file generated in the previous step.

### 3.1. Establishment of Finite Element Model

Figure 9 shows the model of thin-walled parts and the finite element mesh. In this model, the X-axis corresponds to the moving direction of the laser heat source (X-direction), the Z-axis is perpendicular to the moving direction (Z-direction), and Y-axis corresponds to the build direction of thin-walled parts (Y-direction). The size of the 3D model of thin-walled parts is 80 mm × 2.2 mm × 15 mm, the number of cladding layers is 60, the thickness of each layer is 0.3 mm, and the element size of the cladding layer is 2 mm × 0.22 mm × 0.15 mm. In order to reduce the number of grid divisions and unnecessary calculations, a part of the actual substrate is selected for modeling, with a size of 100 mm × 22 mm × 15 mm.

To ensure the accuracy of heat transfer between the thin-walled parts and the substrate, the substrate was cut during mesh division, and the substrate area in contact with the thin-walled parts was mesh-encrypted to make the element size consistent with the cladding layer. The indirect coupling field analysis was used to solve the model. SOLID70 element is selected for the transient thermal analysis of the temperature field. The type Solid70, which was eight-node brick solid elements, has high precision and is suitable for three-dimensional steady-state or transient thermal analysis.

Considering the interaction of various influencing factors in the actual laser cladding process and the complexity of multi-physical field coupling, the finite element model is simplified according to the following principles: (1) Ignoring the influence of reflection, refraction, and other physical phenomena in the laser propagation process, and the performance of the heat source and the generated spot is always unchanged; (2) it is assumed that the effective energy absorbed by the powder and the substrate material is all used for heating up, and the temperature field is only affected by the laser power and thermal property parameters; (3) the influence of molten pool flow on the temperature field and stress field of the cladding layer is not considered; (4) assume that the environmental state is stable and the material is isotropic [14]; (5) the influence of physical and chemical property changes caused by element diffusion in the melting process of powder and substrate material is ignored [15]; and (6) the yield criterion of the substrate and metal powder is the von Mises criterion, and the plastic section of the metal material meets the flow criterion and hardening criterion.

### 3.2. Material Properties of 316L Stainless Steel

The thermal and physical properties of 316L stainless steel at different temperatures are shown in Table 4 [16]. Although the cladding layer and the substrate have differences in crystal grains and microstructures in the metallurgical process, the same physical parameters for the cladding layer and the substrate are used in this paper.

The reason is that studies by Deshpande et al. [17] show that the effect of this method on the results is negligible, and the convergence of finite element calculation can be improved by adopting the same physical parameters of the cladding layer and substrate. Meanwhile, the mechanical properties of additive manufactured 316L stainless steel parts in actual production are anisotropic [18,19]. However, the main parameter data of 316L anisotropic material is difficult to obtain, and the thermodynamic simulation of this model adopts anisotropic simplification processing [20].

### 3.3. Solution and Analysis of Temperature Field of Thin-Walled Parts in AM

#### 3.3.1. Gauss Moving Heat Source Model and Birth and Dead Element

In the present research, ANSYS Mechanical APDL19.2 solver was used to solve the temperature field of thin-walled parts in additive manufacturing. ANSYS Program Designed Language (APDL) was used to implement all of those laser cladding procedures. The moving heat source and the setup of the birth and death elements described below are implemented in APDL.

Moving heat source and “birth and death elements” are two key settings to simulate the laser cladding process. The moving heat source is used to simulate the laser cladding process by defining the heat flux function and loading it onto the pre-defined cladding surface step by step. The cladding layer formation is simulated using the “birth and death element” technology provided by ANSYS. Before loading the heat source for calculation, all cladding layer elements are set to “death’’, and then according to the time parameters and load steps, the “dead” elements are activated gradually through the cyclic accumulation of X-axis coordinate values. After the “born” elements are incorporated into the calculation model, so as to realize the simulation of the gradual growth process of the cladding layer along with the laser spot movement. At the same time, heat flux is applied to the surface of these “born” elements to achieve heat conduction.

Gaussian heat source model mainly includes Gaussian surface heat source and Gaussian body heat source, among which the Gaussian surface heat source is relatively widely used. Considering that in the laser cladding process, the thickness of the powder flow converging under the laser spot is relatively thin, the laser energy can be equivalent to a two-dimensional surface heat source [21]:(1)$$I\left(r,w\right)=\frac{2AP}{\pi \omega^{2}}exp\left(-\frac{2r^{2}}{\omega^{2}}\right)$$
(2)$$r=\sqrt{{\left(x-x_{0}\right)}^{2}+{\left(y-y_{0}\right)}^{2}}$$
where, *A* is the laser energy absorption rate of the material, where the value is 0.38 [22]; *P* is the laser power of the heat source; *ω* is the laser radius; *r* is the characteristic laser radius, and ($x_{0}$,$y_{0}$) is the initial position of the heat source.

#### 3.3.2. Boundary Conditions

In the numerical simulation process, corresponding initial conditions and boundary conditions must be given to obtaining the unique solution of the differential equations. The principle of setting the initial conditions and boundary conditions is to simulate the actual laser additive manufacturing experiment process as much as possible. The initial conditions and boundary conditions of numerical simulation of laser additive manufacturing process in this paper are set as follows:(1)Initial conditions

In the simulation process of additive manufacturing of thin-walled parts, the initial temperature of the substrate and the ambient temperature were set as room temperature 22 °C at the initial time.
(2)Boundary conditions

In the AM process of thin-walled parts, the surface of the cladding layer in contact with the air dissipates heat to the surrounding air mainly through convection and radiation, which is the third boundary condition. If the convection and radiation are calculated separately, it will increase the nonlinearity of the temperature field and reduce the convergence of the calculation process. Therefore, in order to ensure the convergence of the temperature field in the laser additive manufacturing process, the convection and radiation heat transfer modes were combined and adopted a hybrid heat transfer coefficient [23]. The derivation process is as follows:(3)$$\alpha_{t}=\varepsilon \sigma \left(T+T_{0}\right)\left(T^{2}+T_{0}^{2}\right)+h_{conv}\;$$
(4)$$q_{t}=\alpha_{t}\left(T-T_{0}\right)$$
where, *h_conv_* is the convective heat transfer coefficient of the protective gas, which is about 10 W/(m^2^·°C). *T* is the boundary surface temperature of the object; *T_0_* is the ambient medium temperature; *ε* is the emissivity of the 316L stainless steel plate; *σ* is the Steffen Boltzmann constant, about 5.68 × 10^−8^ W/(m^2^·°C^4^), *α_t_* is the hybrid heat transfer coefficient related to temperature, and *q_t_* is the total heat at the surface. The “Film Coefficient” under the Convection option in Workbench can be defined by “Function”, so that both convection and radiation can be calculated. The surface in contact with the cladding channel was excluded from the substrate, and the heat transfer coefficient was applied to its outer surface.

The lower surface of the cladding substrate is in contact with the fixture. In practical application, heat transfer is realized by heat conduction, and the heat dissipation formula of heat conduction is:(5)$$q_{cond}=-k\frac{dT}{dx}$$
where, *k* is the thermal conductivity of the contact part between the bottom surface of the substrate and the fixture. In order to simplify the simulation, the fixture was not modeled, but the convection on the bottom surface of the substrate was considered. Equation (3) was used to calculate the heat dissipation of the bottom surface of the substrate, in order to simplify the calculation, where α_t_ was set at 60 W/(m^2^·°C).

#### 3.3.3. Treatment of Latent Heat

In the process of laser cladding deposition, the metal powders absorb heat and melt from the powder state into a liquid form. When the laser heat source is swept over, the liquid metal in the molten pool releases heat and solidifies into a solid. In this process, there are also heat conduction, heat radiation, and other phenomena. The solidification cooling of the cladding layer is related to the absorption and release of latent heat.

Therefore, in order to make the simulation results more accurate and more consistent with the actual process of metal additive manufacturing, it is necessary to consider the phase transition caused by the melting and solidification of powder in the numerical simulation of the temperature field.

The equivalent specific heat capacity of 316L stainless steel powder is shown in Figure 10. It can be known that the phase transition region of 316L stainless steel is (1650–1723 K), and the latent heat of the phase transition is up to 270 kJ/kg, so the phase transition cannot be ignored.

In ANSYS, the latent heat is processed by the enthalpy method, that is, the enthalpy calculates the latent heat at different temperatures. The product of density and specific heat is integrated into temperature. The calculation formula of enthalpy is as follows:(6)H=∫ρcdT
where, *H* is enthalpy, *ρ* is the density of the material, *c* is the specific heat capacity of the material, and *T* is the temperature.

#### 3.3.4. Solution and Analysis of Temperature Field

After initial conditions and boundary conditions are set, the solution of temperature field was realized according to the following process (Figure 11):

The simulation results of temperature field of 316L stainless steel thin-walled parts formed by laser cladding are as follows (Figure 12):

Figure 12a shows the process of applying heat flux and loading moving heat source on the first cladding layer. Figure 12b is the solution result of the temperature of the thin-walled parts at the end of the cladding. The result shows that the temperature of the laser spot position reached 3258.36 °C at the end of the processing. The lowest temperature also reached 707.552 °C.

Figure 13 shows the comparison between the temperature field of thin-walled parts solved by ANSYS finite element at a certain moment and recorded by the infrared camera. The results show that the temperature field of thin-walled parts simulated by ANSYS is basically consistent with the temperature distribution of thin-walled parts in actual additive processing. Figure 14a shows the temperature change curve of the node at the center point on the surface of the thin-walled parts (marked with the red dot in Figure 14a, which is the same as the position of the cross cursor in Figure 8) extracted after ANSYS solution. It can be seen that the highest temperature experienced by this point is 2119.22 °C. The temperature history recorded by the infrared camera in the same time period was compared with the temperature field results of finite element simulation, as shown in Figure 14.

Compared with the simulation process of some researchers before, we have made some efforts to reduce the error as much as possible from the following aspects: (1) When establishing the finite element model in ANSYS, the accuracy of the heat transfer between the thin-walled part and the substrate was considered. Moreover, we made sure that the grid of the thin-walled part and the grid of the substrate are connected together when dividing the grid; (2) with reference to the interaction between the laser and the metal powder in the actual machining process, the heat source model used for the temperature field calculation was determined to be the modified Gaussian surface heat source model, which takes into account the material’s absorption rate of laser energy; (3) flow fluid in the molten pool was not considered in this study in order to simplify the calculation process, but the thermal conductivity of the molten state was enhanced to account for convection heat transfer in the molten pool; and (4) especially, the latent heat of phase change caused by the melting and solidification of metal powder in the numerical simulation of the temperature field was considered and processed.

However, there is discrepancies between calculated and measured results (temperature difference, mainly for the following reasons: (1) Compared with the actual laser cladding process, the finite element calculation model is based on some simplification, especially the setting of boundary conditions is quite different from the actual process; (2) the thermal property parameters of 316L stainless steel used in the finite element calculation model are not completely consistent with that used in the machining experiment, and some of the parameters are obtained by extrapolation method, which has some errors; and (3) the infrared thermal imaging camera used a fixed emissivity when monitoring the thin-walled parts’ temperature in the AM process, while the emissivity of the thin-walled parts during the processing is constantly changing with its temperature changes.

Therefore, comparing the calculated temperature field results of ANSYS finite element with the monitoring results of the infrared camera, the simulation results of temperature field within the allowable error range have reference significance. Furthermore, it can solve the thermal stress distribution and deformation to guide the subsequent milling.

### 3.4. Solution and Analysis of Stress Distribution and Deformation of Thin-Walled Parts in AM

#### 3.4.1. Element Conversion and Boundary Conditions


(1)Element conversion


The calculation of the stress field in the AM of thin-walled parts was carried out on the temperature field, and the geometric model and mesh division was consistent with the setting of the temperature field. However, before the calculation of the stress field, the element type needs to be converted by ETCHG command, which can convert the SOLID70 thermal element set when solving the temperature field above into a structural element. The default conversion structural element here was SOLID185 element, which is defined by eight nodes, each of which has three degrees of freedom: translation in node X, Y, and Z directions. The element has plasticity, super-elasticity, stress stiffening, creep, large deflection, and large strain capacity.
(2)Boundary conditions

The boundary conditions of the stress field need to consider the actual situation of the thin-walled parts in the AM process. In AM and cooling of thin-walled parts, the substrate was permanently mounted on the fixture. The front and back surfaces and the bottom surfaces of the substrate can be regarded as having fixed constraints, while the remaining surfaces have no constraints. The whole LMD process includes laser cladding stage and subsequent cooling stage. Accordingly, different boundary conditions are considered in the two stages. Specifically, the Film Coefficient of the cooling stage is set as “Stagnant Air—Horizontal Cycle” under the Temperature Dependent dissipation option. In the process of solving the stress field, a schematic diagram of applying fixed constraints to the substrate is shown in Figure 15.

#### 3.4.2. Results and Analysis of Thermal Stress Distribution and Deformation of Thin-Walled Parts

In calculating the stress field, to facilitate calculation and save memory space, the ANSYS + Workbench co-simulation method was used to calculate the stress field of thin-walled parts manufactured by laser cladding. In the mechanical analysis of Workbench, Hook’s law with temperature-dependent Young’s modulus and Poisson’s ratio was employed to model the elastic strain. For the plastic strain, the von Mises yield criteria was employed as a yield point function. A bilinear kinematic hardening principle was employed to model the plastic hardening behavior of the additive thin-walled part is simulated when it is in a yielding state. The yield stress strength of 316L stainless steel was set at 550 MPa, which was identical to the experiment measurement. The Newton–Rapson method and the linear search option were activated in the calculation process, and the large deformation option was turned on to get a better convergence solution.

The von Mises stress results obtained by Workbench are shown in Figure 16a,b. In terms of stress distribution, the maximum stress appears at both ends (about 3.6 mm) of the thin-walled parts when the cladding reaches a certain height after the AM. According to the normal stress results in Figure 16c, the stress is tensile stress between adjacent cladding layers. The maximum stress occurs next at the junction between the bottom of the thin-walled parts and the substrate. According to the normal stress results in Figure 16d, it is compressive stress. The thermal stress distribution is related to the path of the laser heat source, and thermal stress presents gradient distribution in thin-walled parts. The maximum value of thermal stress is 316.27 MPa before cooling.

For thermally stressed thin-walled parts after the addition is completed, the maximum deformation appears at the two ends of the cladding layer (about 3.3–7.8 mm) at a certain height from the substrate, which coincides with the position where the maximum stress occurs. Moreover, the second deformation position is the top ends of the thin-walled parts, as shown in Figure 17a,b. It should be noted that from the normal stress contours, these two locations are just the mixing zone and transition zone of tensile and compressive stresses. The thermal stress deformation of the thin-walled parts shows asymmetrical distribution along the laser cladding direction. The thermal deformation fluctuation at the top of thin-walled parts is the most dramatic, and the thermal deformation at the bottom is the least. The maximum deformation of the thin-walled part after AM is 0.183 mm. It can be seen that there is a strong correlation between the residual stress and deformation of the AM parts. The difference value of tensile and compressive residual stresses in the thin-walled parts is the main reason for the depression deformation in the lower part and the bulge deformation in the upper part.

## 4. Milling Experiments of 316L Stainless Steel Parts Manufactured by AM

### 4.1. Side Milling Experiments of Thin-Walled Parts Manufactured by AM

#### 4.1.1. Experiment Condition

The milling tool used in this experiment is an integral carbide milling cutter GM-4E-D10.0 with TiAlN coating. The helix Angle is 45°, and the blade length is 25 mm. Refer to the recommended stainless steel milling cutting parameters, the milling parameters are set as: spindle speed n = 2200 r/min, feed speed f = 135 mm/min, maximum cutting depth a_e_ = 0.1D = 1 mm, a_p_ = 1.5D = 15 mm.

The milling experiments of thin-walled parts were carried out with the cutting parameters recommended by milling cutters. As the surfaces of the additive manufactured thin-walled parts were very rough, and there was a hardening layer, the working engagement of the cutting edge a_e_ = 0.5 mm and the back engagement of the cutting edge a_p_ = 3 mm were selected to perform side milling on the surface of the thin-walled parts. Meanwhile, the KISTLER three-way dynamometer was used to measure the milling force during the milling process. Figure 18 shows the side milling and milling thin-walled parts.

#### 4.1.2. Experiment Results and Analysis

After filtering and denoising the data measured by the dynamometer, the three directional forces were calculated as F_x_ = 382.927 N, F_y_ = 198.4 N, and F_z_ = 53.422 N, respectively. The surface roughness of the thin-walled parts after milling was randomly sampled by a hand-held surface roughness measuring instrument, and the measurement results showed that the minimum surface roughness reached Ra = 0.320 ± 0.1 μm, indicating that the surface quality of the laser cladding thin-walled parts was significantly improved after milling.

### 4.2. Side Milling Orthogonal Test of Block Specimens Manufactured by AM

For thin-walled parts, the proportion of side milling should be more significant than that of end milling in the actual machining process due to the feature of ASHM. According to the simulation results and the actual AM process of thin-walled parts showed the morphology of wide at the top and narrow at the bottom, dynamic compensation of cutting force is needed to obtain high dimensional accuracy of thin-walled parts after milling. As the mechanical properties of 316L stainless steel additive manufactured parts are different from those of traditional forged parts, the relationship between milling parameters and milling forces of forged parts cannot be applied to the additive manufactured parts. Therefore, a three-factor and four-level orthogonal test L_16_(4^3^) was designed for the side milling of 316L stainless steel additive manufactured part. The relationship between the milling parameters of the additive manufactured part and the milling force was studied. At the same time, to improve the test efficiency, save the test materials, and shorten the test time, the cube-shaped specimen was used as the test object for the side milling test. Four groups of side milling tests can be performed on four sides of a cube-shaped specimen. The orthogonal test results of side milling of additive manufactured block specimen are shown in Figure 19. The parameters of factor level in the orthogonal milling test are shown in Table 5, and the orthogonal test design and test results are shown in Table 6.

The empirical formula for calculating cutting force is $F_{c}=Ca_{e}^{k}f^{m}v_{c}^{n}$, according to the data in Table 5, the regress function in MATLAB was used to obtain the regression equation coefficient, residual, and statistics for testing the regression model with the data of each variable. The coefficient of the regression equation is calculated as follows: $\{\begin{array}{c} C=313.2495\\ k=0.37728\\ m=0.26648\\ n=-0.17785 \end{array}$. Additionally, the empirical formula for calculating cutting force during side milling of 316L stainless steel additive manufactured parts was obtained as $F_{c}=313.2495a_{e}^{0.37728}f^{0.26648}v_{c}^{-0.17785}$. When using regress function to carry out multiple regression analysis and get the coefficient of the regression equation, the linear correlation coefficient *R^2^* and the significance probability value *p* of variance analysis can be obtained from stats, the statistic used to test the regression model, where *R^2^* = 0.968, *p* = 0.31245 × 10^−8^. It shows that the regression equation can fit the data well.

## 5. Milling Simulation and Deformation Prediction of 316L Stainless Steel Additive Manufactured Thin-Walled Parts

After solving the thermal stress field of thin-walled parts, the results were saved, and new structure analysis was built. Based on the solving result of the temperature field, the “birth and death elements” technology was used again to simulate the milling process. The milling force measured in Section 4.1.2 above was loaded to the node of the subtracted element to be milling in the form of a three-way force. With the gradual movement of the three-way force, the previous “living element” becomes “dead element” again, indicating that the subtracted element is separated from the matrix. The schematic diagram of simulating the milling process with the “birth-and-death elements” technology is as follows (Figure 20):

The stress distribution and deformation simulation results of the additive manufactured thin-walled parts after side milling are shown in Figure 21. It can be seen that the milling force introduced by milling obviously changed the stress distribution and size of the thin-walled parts manufactured by AM. In terms of stress distribution, the maximum residual stress on the thin-walled parts after milling was transferred from the two ends of the thin-walled parts to the thin wall and the substrate connection and the substrate assembly position. The stress distribution on the thin-walled parts showed regional symmetry with the movement of the milling cutter.

The maximum value of von Mises residual stress in the thin-walled parts after milling was 2570 MPa. Moreover, the residual stress generated after milling of thin-walled part is mainly compressive from the third principal stress contours and Y component (cladding layer accumulation direction) stress contours (Figure 22), and milling introduced large compressive stress, about 418 MPa.

Therefore, the milling process can release the tensile stress on the surface of thin-walled parts and improve the friction resistance and fatigue strength of the surface of thin-walled parts. Combined with the subsequent post-treatment, it will be more beneficial to improve the life of the thin-walled parts. In addition, it can be seen from Figure 21 that milling reduced the thermal stress deformation of the additive manufactured thin-walled parts, but does not cause significant additional deformation (the maximum deformation is 0.122 mm). This indicates that the additive manufactured parts’ forming accuracy and surface quality can be improved obviously by subtractive processing.

In order to verify the accuracy of the finite element model of the milling of the additive manufactured thin-walled parts, after the side milling, the laser distance sensor was used to scan the top of the thin-walled parts, and the deformation data recorded is shown in Figure 23a. The deformation of thin-wall parts has been improved obviously after side milling. As shown in Figure 23b, the maximum deformation on the top of the laser cladding thin-wall parts decreases from 1.412 mm to 0.280 mm after milling. The deformation fluctuation range reduces significantly.

For the deformation after milling, we compared the values solved by the finite element model with those recorded by the laser distance sensor. After deleting the outliers, we filtered and denoised the values recorded by the laser distance sensor. As shown in Figure 24, compared with the deformation value measured by the laser distance sensor, the deformation value predicted by the finite element model is relatively small, mainly for the following reasons: (1) the boundary conditions of the finite element model in the solution process are correspondingly simplified compared with the actual processing process of the additive manufactured thin-walled parts; (2) ANSYS simulation milling is an implicit dynamic simulation, which is in the form of applying a three-way force to simulate milling, which is different from the actual process of milling cutter cutting thin-walled parts, leading to some errors in the solution results; (3) the mesh division of the finite element model and the setting of the solution load step will have a particular impact on the accuracy of the solution results; and (4) the finite element simulation results show that the deformation of thin-walled parts is plastic primarily deformation, and the measurement of plastic deformation by the laser distance sensor is affected by the measurement time, so there will be errors between the two.

## 6. Discussion

In the process of metal cutting, different stress field distribution has a significant influence on machining deformation. In the whole process of ASHM, for the AM process, the stress field distribution of thin-walled parts is mainly affected by the temperature field; for milling processing, the stress field of thin-walled parts is mainly affected by the cutting process parameters or cutting force. Miguelez et al. [24] found that the residual stress of the machined surface was affected by the combination of processing thermal load and mechanical load through simulation analysis and experimental verification. B. Li et al. [25] studied the effect of cutting depth on the distribution of residual stress and the thin-walled structure of deformed aluminum alloy and pointed out that it can reduce the residual stress of the workpiece by controlling the processing parameters, reduce the deformation of the thin-walled structure and improve the machining accuracy. Okushima and Kakino pointed out that when the surface of the workpiece is affected by a thermal load, residual tensile stress is generally generated, and when the surface of the workpiece is affected by a mechanical load, residual compressive stress is generally generated [26].

After the AM, it can be seen from the normal stress contour that the residual stress at the top of thin-walled parts was symmetrically distributed on the central axis. It indicates that the tensile stress at both ends gradually changes into compressive stress during the transition from both ends to the middle. The value of normal stress decreases first and then increases. After milling, it can be seen from the third principal stress contour that the residual stress distribution at the top of the thin-walled parts was similar to that after the AM. Compared with the residual stress after AM, the tensile stress at both ends after milling was reduced by about 94.5%, and the compressive stress in the middle was increased by about 49%. The tensile stress at both ends was much less than the compressive stress in the middle. Therefore, in addition to releasing the tensile stress, milling also introduced additional compressive stress, which changed the distribution of the additive’s stress field and led to the deformation of the thin-walled parts. It can be seen that, for thin-walled parts manufactured by LMD, the subsequent milling processing can achieve active control of near-surface stress, achieving the purpose of improving part performance and production efficiency.

It is worth thinking about how the temperature field and stress field inside the part will change the microstructure of the part in the process of ASHM. The microstructure has a great influence on the performance of the material. Further observation and comparison of the internal microstructure of thin-walled parts under different additive and subtractive process parameters will help to further understand how to improve the forming quality of thin-walled parts processed by ASHM.

In addition, in the actual processing of LMD, the temperature gradient in the molten pool leads to a surface tension gradient, which is related to Marangoni convection. Some researchers have found that it will affect the continuity and stability of the scan trace during LMD processing [27]. This is also an important factor worthy of follow-up research and verification.

## 7. Conclusions and Summary

The additive and subtractive manufacturing thin-walled parts were carried out with the laser cladding composite processing equipment built. At the same time, the finite element model of laser cladding and milling thin-walled parts was established, and the temperature field and stress field of laser cladding and milling process were numerically simulated by using ANSYS “birth-and-death elements” technology.

For the temperature field, a point is selected on the thin-walled part’s outer surface, and the time-temperature curve of the point was obtained. The temperature rising process has a gradually increasing peak before the arrival of the laser heat source. After the heat source, the temperature peak decreases, and the temperature fluctuation gradually tends to be stable. The temperature dropping curve is approximately one of the inverse proportional function curves.

After the AM, the stress distribution in the thin-walled parts is different along with different directions, and stress concentration and stress mutation can be formed at the cladding layer and substrate joint. The tensile stress concentration is the most serious at both ends of thin-walled parts at a certain height away from the substrate. The stress deformation at this place and experiment results also prove this conclusion.

After milling, the distribution and value of the residual stress in thin-walled parts changed significantly. Due to the compressive stress introduced in milling, the tensile stress introduced by temperature gradient changes to compressive stress after the AM of thin-walled parts. The stress field is no more extended gradient distribution, but regional distribution with the movement of milling force.

For deformation caused by thermal stress, milling the thin-walled parts at the end of the AM can reduce the thermal stress. The experimental results are in good agreement with the simulation results. Still, there is a specific deviation between the two values, which can be improved by improving the thermal property parameters of 316L stainless steel powder and applying more accurate measurement methods.

## Figures and Tables

**Figure 1 materials-14-05582-f001:**
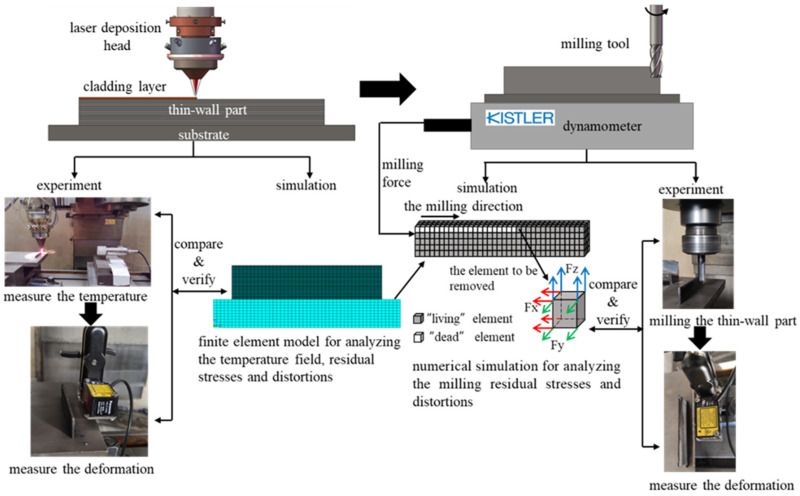
The research content structure of this paper.

**Figure 2 materials-14-05582-f002:**
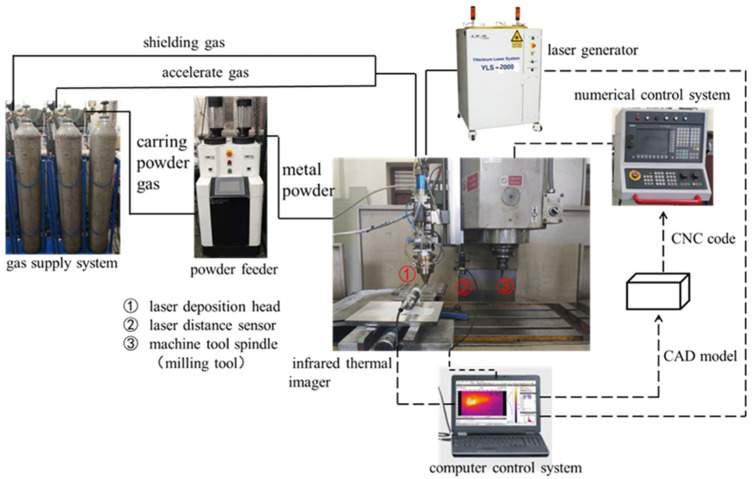
Experimental system of laser cladding additive and subtractive hybrid manufacturing.

**Figure 3 materials-14-05582-f003:**
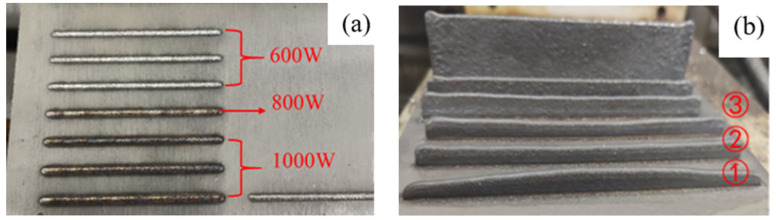
Single-pass cladding layer forming experiment: (**a**) single-layer; (**b**) multilayers.

**Figure 4 materials-14-05582-f004:**
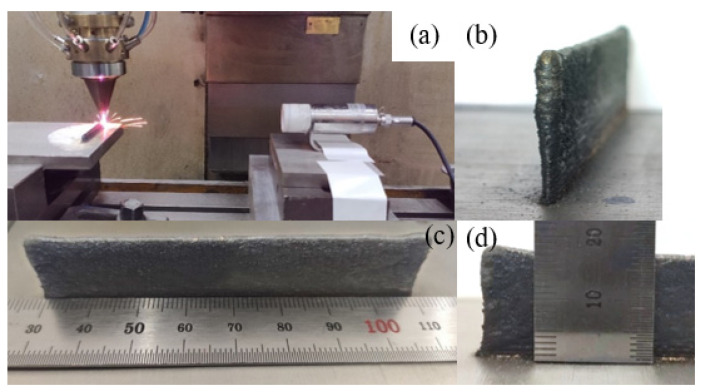
The 316L stainless steel thin-walled parts formed by laser cladding: (**a**) Processing process; (**b**) side morphology; (**c**) dimensions of thin-walled parts; (**d**) morphology of one end of the thin-walled parts.

**Figure 5 materials-14-05582-f005:**
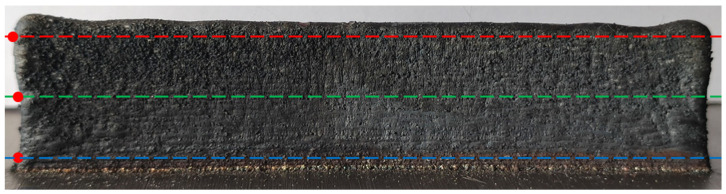
Scanning position and scanning reference point of the laser distance sensor.

**Figure 6 materials-14-05582-f006:**
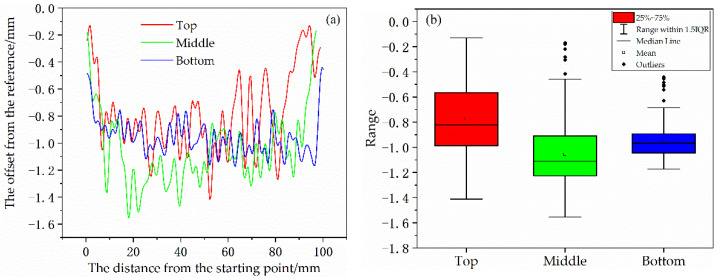
Deformation (**a**) and deformation statistics (**b**) of different positions of thin-walled parts.

**Figure 7 materials-14-05582-f007:**
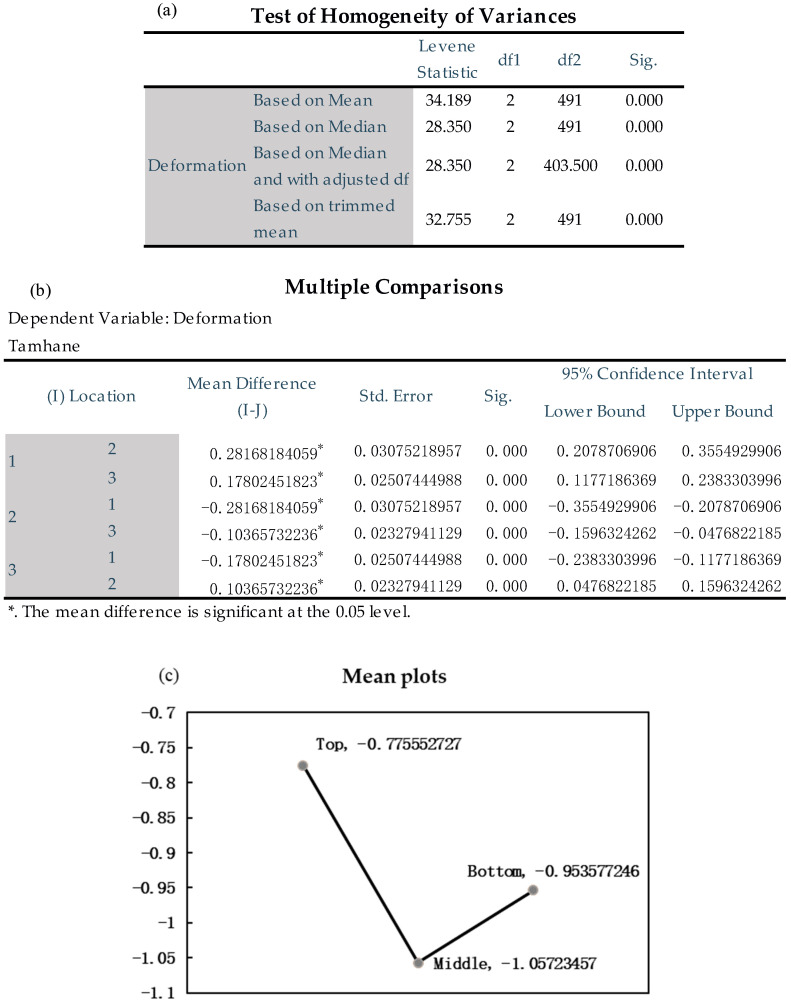
The results of one-way ANOVA analysis and post hoc multiple comparisons: (**a**) test of homogeneity of variances; (**b**) post hoc tests; (**c**) means plots.

**Figure 8 materials-14-05582-f008:**
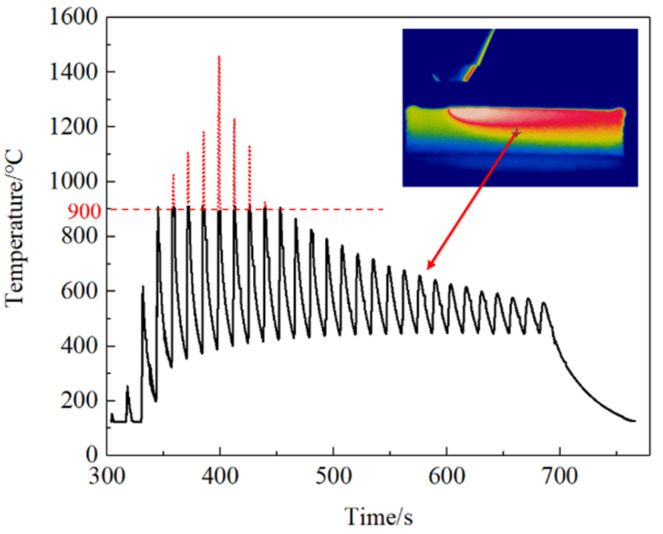
A point’s time-temperature history recorded by the infrared thermal imager.

**Figure 9 materials-14-05582-f009:**
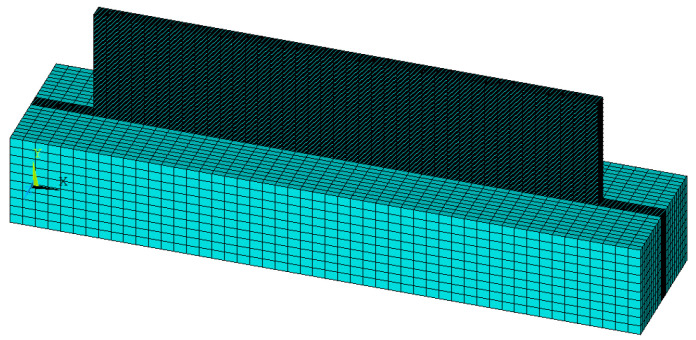
Finite element model and meshing of thin-walled (with substrate).

**Figure 10 materials-14-05582-f010:**
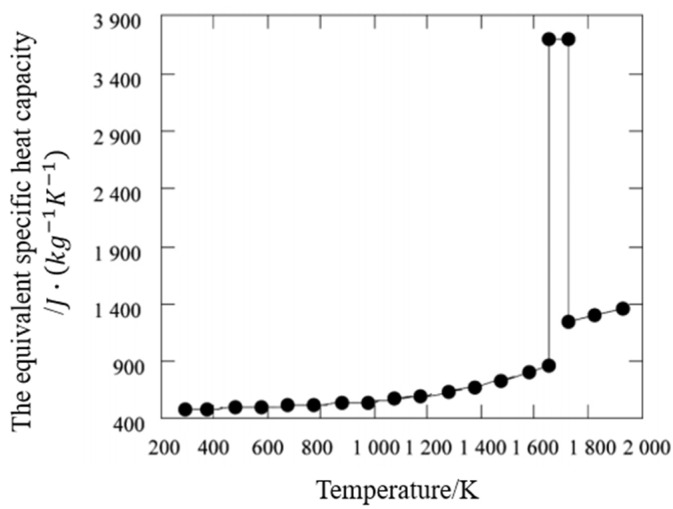
Equivalent specific heat capacity of 316L stainless steel powder at different temperatures.

**Figure 11 materials-14-05582-f011:**
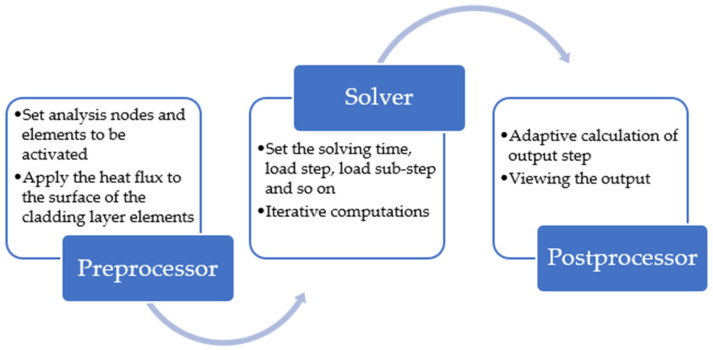
Finite element method for solving temperature field of thin-walled parts manufactured by AM.

**Figure 12 materials-14-05582-f012:**
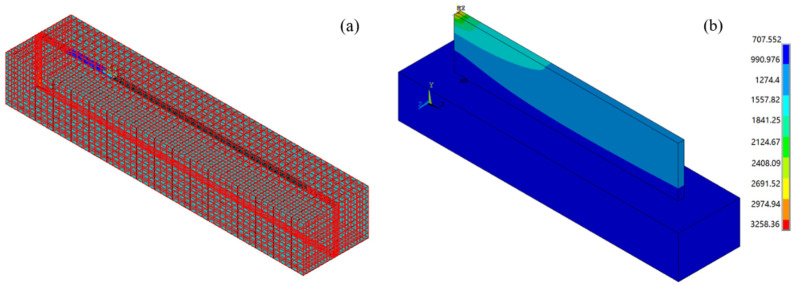
Finite element solution of thin-walled parts formed by laser cladding: (**a**) applying heat flux and loading moving heat source to the first cladding layer; and (**b**) the solution result of temperature field of thin-walled parts at the end of cladding.

**Figure 13 materials-14-05582-f013:**
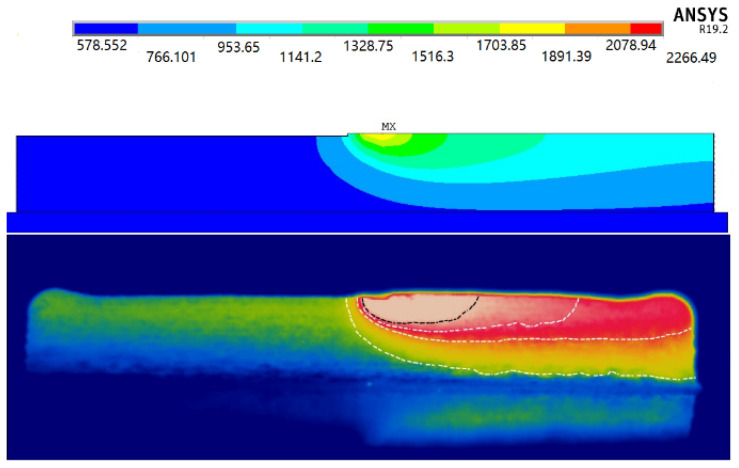
Temperature field distribution of thin-walled parts at a specific time: temperature distribution of thin-walled parts solved by ANSYS (**top**); and temperature distribution of thin-walled parts recorded by the infrared camera (**bottom**).

**Figure 14 materials-14-05582-f014:**
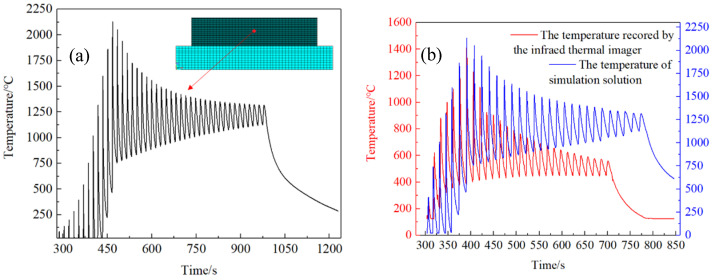
(**a**) Temperature-time history of the surface center point in the AM process of thin-walled parts solved by ANSYS. (**b**) Comparison of temperature experiment results and simulation results of 316L stainless steel thin-walled parts formed by laser cladding.

**Figure 15 materials-14-05582-f015:**
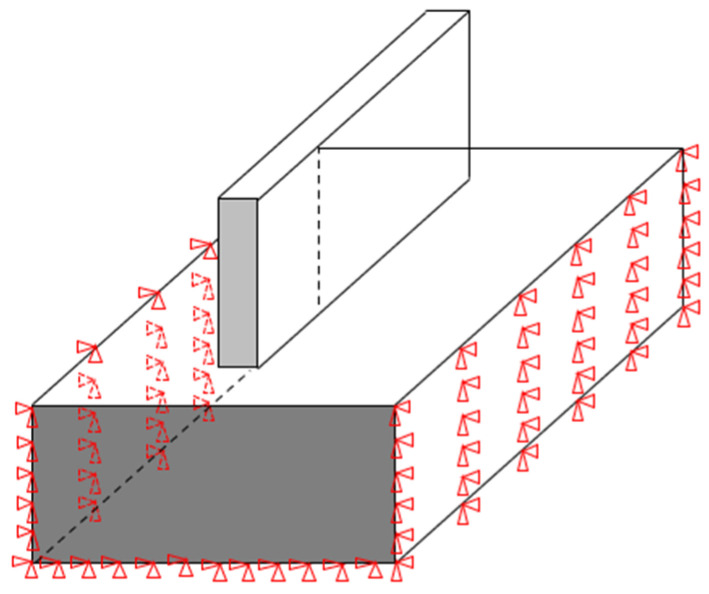
Schematic diagram of setting fixed constraints to the substrate for milling thin-walled parts.

**Figure 16 materials-14-05582-f016:**
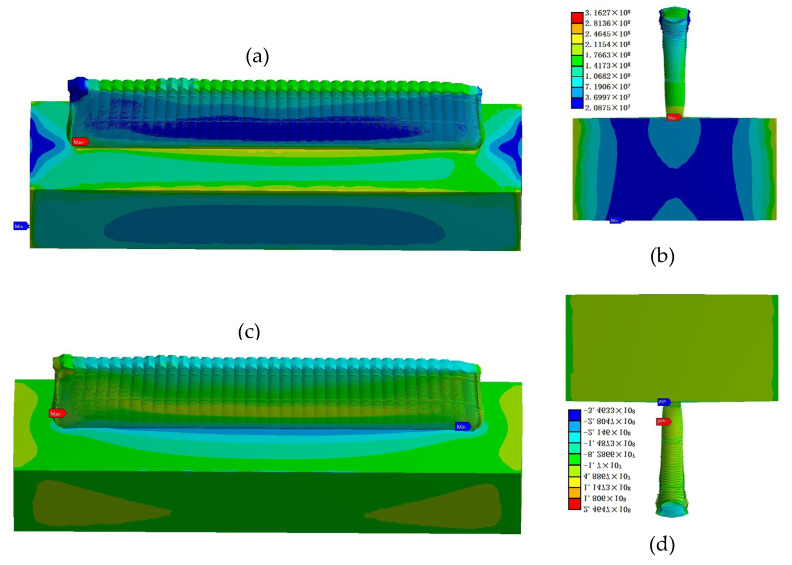
Finite element analysis results of stress field of thin-walled parts at the end of AM (amplification: 17×): (**a**,**b**) Von Mises stress distribution; (**c**,**d**) normal stress distribution.

**Figure 17 materials-14-05582-f017:**
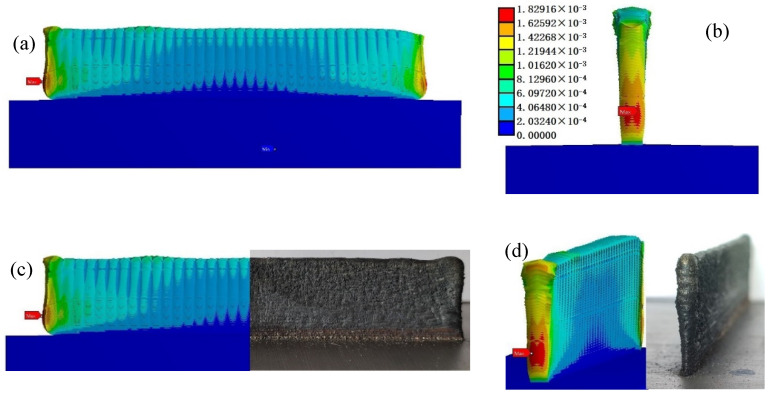
Finite element solution results of stress and deformation of the thin-walled parts (amplification: 17×): (**a**) surface deformation; (**b**) deformation at both ends; (**c**,**d**) comparison between simulation and actual deformation of thin-walled part.

**Figure 18 materials-14-05582-f018:**
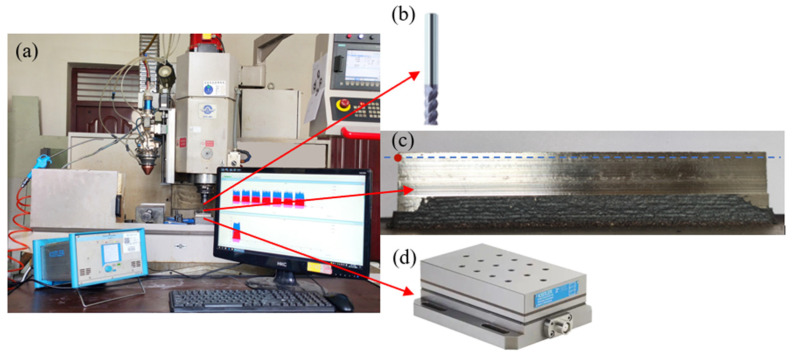
Side milling and force measurement of laser cladding thin-walled parts: (**a**) milling force measurement system; (**b**) GM-4E end mill; (**c**) thin-walled parts after milling; and (**d**) KISTLER triaxial dynamometer.

**Figure 19 materials-14-05582-f019:**
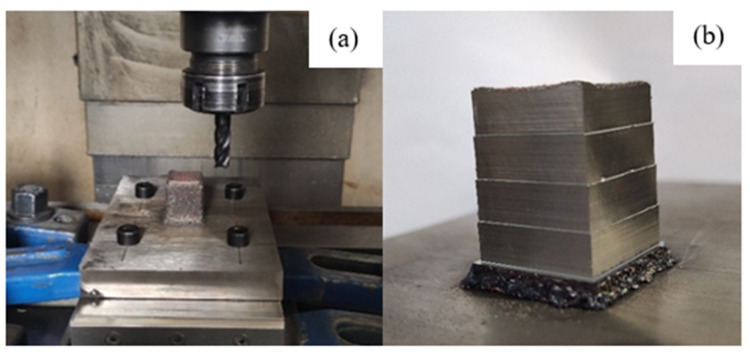
(**a**) Side milling for the AM block specimen; and (**b**) block specimen after orthogonal side milling test.

**Figure 20 materials-14-05582-f020:**
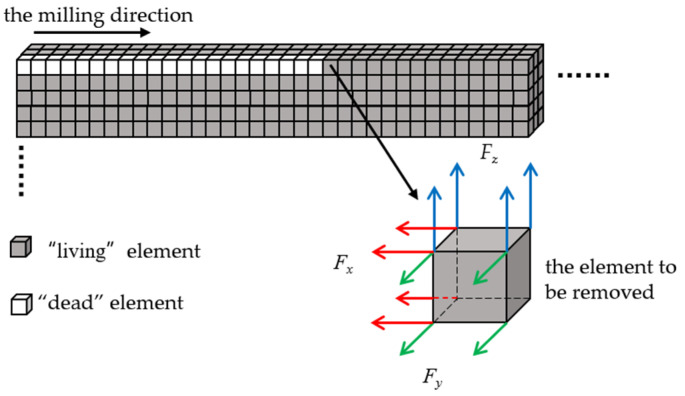
Simulation of the milling process with “birth-and-death element” technology.

**Figure 21 materials-14-05582-f021:**
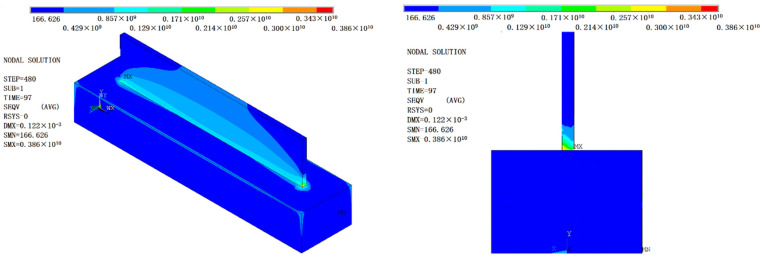
Von Mises residual stress distribution and deformation of thin-walled parts after side milling.

**Figure 22 materials-14-05582-f022:**
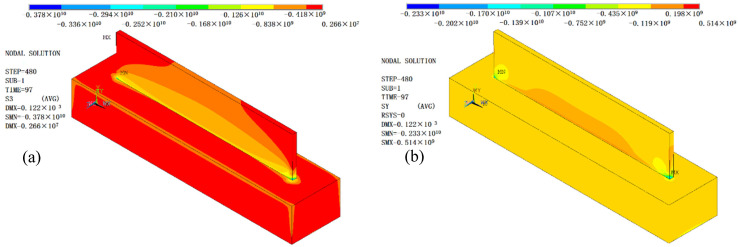
(**a**) The third principal stress contours after side milling of thin-walled parts; and (**b**) Y component normal stress contour after side milling of thin-walled parts.

**Figure 23 materials-14-05582-f023:**
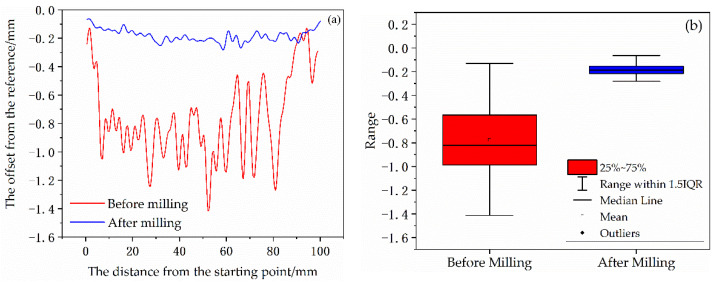
Comparison of deformation (**a**) and deformation statistics (**b**) before and after milling on the top of laser cladding thin-walled parts.

**Figure 24 materials-14-05582-f024:**
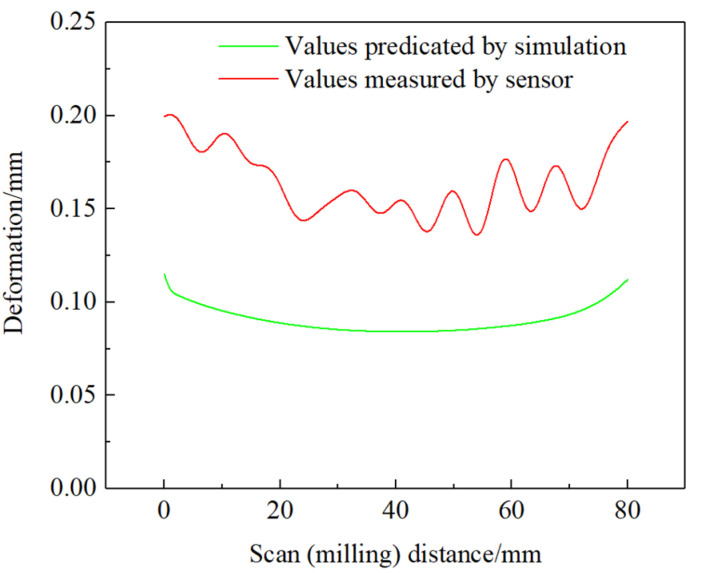
Comparison of the predicted and measured deformation of thin-walled parts after milling.

**Table 1 materials-14-05582-t001:** Chemical components of 316L stainless steel powder.

Chemical Element Components	C	Si	Mn	P	S	O	Ni	Cr	Mo
(wt%)	0.025	0.53	1.46	0.01	-	0.53	11.78	16.69	2.41

**Table 2 materials-14-05582-t002:** Laser cladding test parameters and values.

Equipment	Adjustable Parameters	Value
Laser generator	Laser power (W)	1000, 800, 600
Powder feeder	Powder feed rate (g/min)	6.271
Gas supply system	Carrier gas pressure (MPa)	3.5
	Shielding gas pressure (MPa)	3.5
Laser deposition head	Spot diameter (mm)	2
Machine tool	*Z*-axis lifting (mm)	0.5, 0.4, 0.3
	Scanning interval time between layers (s)	20, 10, 0

**Table 3 materials-14-05582-t003:** Laser cladding parameters of 316L stainless steel thin-walled parts.

Parameters	Value
Laser power (W)	600 (520)
Scanning speed (mm/s)	6
Powder feed rate (g/min)	6.271
*Z*-axis lifting (mm)	0.3
Scanning interval time between layers (s)	0
Cladding length (mm)	80
Cladding width (mm)	2.2
The number of clad layers	50

**Table 4 materials-14-05582-t004:** Thermal and physical properties of 316L stainless steel.

Temperature	Density	Thermal Conductivity	Specific Heat	Poisson’s Ratio	Coefficient of Thermal Expansion	Yield Strength	Young’s Modulus
℃	$kg/m^{3}$	W/(m · K)	J/(kg · K)		${℃}^{-1}$	MPa	GPa
10	7740	13.1	450	0.29	17.62 × 10^−6^	249	200
100	7710	14	500	0.29	17.87 × 10^−6^	235	188
300	7680	23	650	0.31	18.2 × 10^−6^	221	170
500	7580	36	750	0.32	19.34 × 10^−6^	210	164
800	7470	38	850	0.32	20.86 × 10^−6^	120	131
1200	7350	35	780	0.32	21.61 × 10^−6^	40	59.5
1500	7000	70	920	0.32	25.3 × 10^−6^	0.003	4.56

**Table 5 materials-14-05582-t005:** Parameters of factor level in milling orthogonal test.

The Milling Parameters	Level 1	Level 2	Level 3	Level 4
Milling speed/($m\;\cdot \;min^{-1}$)	55.292	62.204	69.115	76.026
Feed speed/($mm\;\cdot \;min^{-1}$)	108	121.5	135	148.5
$a_{e}$/mm	0.25	0.5	0.75	1

**Table 6 materials-14-05582-t006:** Orthogonal test design and test results of side milling of AM block specimen.

Test Number	Milling Speed $\left(m\;\cdot \;min^{-1}\right)$	Feed Speed $\left(mm\;\cdot \;min^{-1}\right)$	$a_{e}$ (mm)	Milling Force (N)
1	55.292	108	0.25	324.883
2	55.292	121.5	0.75	487.283
3	55.292	135	1	608.572
4	55.292	148.5	0.5	428.174
5	62.204	108	0.5	394.971
6	62.204	121.5	1	523.607
7	62.204	135	0.75	513.708
8	62.204	148.5	0.25	324.572
9	69.115	108	0.75	458.388
10	69.115	121.5	0.25	302.808
11	69.115	135	0.5	434.568
12	69.115	148.5	1	565.371
13	76.026	108	1	497.724
14	76.026	121.5	0.5	404.035
15	76.026	135	0.25	340.161
16	76.026	148.5	0.75	472.565

## Data Availability

The data that support the findings of this study are available from the corresponding author upon reasonable request. The data are not publicly available due to privacy or ethical restricitions.

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
