# Peer review of "Deformation Prediction and Experimental Study of 316L Stainless Steel Thin-Walled Parts Processed by Additive-Subtractive Hybrid Manufacturing"

_materials, 2021, doi:10.3390/ma14195582_

Round 1
Reviewer 1 Report
I have the following comments and suggestion for the authors:
Abstact:
- Please clearly indicate the purpose of your study.
- Please use spaces between numbers and units. Please correct that in the entire manuscript.
Introduction:
- This section is very superficial. There are dozens of papers which deal with the similar problems, i.e. machining of additively manufactured parts. Here are some recent examples to consider:
https://doi.org/10.1007/s00170-020-06387-3
10.3390/ma14112794
http://dx.doi.org/10.3390/jmmp4020032
etc.
2. Based on your review, I do not know what research gap your paper fills. What is the rationale behind your paper. The literature review should have the synthesis which is base of your motivation to do your research.
Materials and methods:
- Please indicate the chemical composition of your powder (could in a form of a table).
- What are the statistical differences between results and presented in figure 6b - p-value for ANOVA or other applicable test.
- What is the positioning accuracy and repeatability of cartesian positioning system you use in your process. This can seriously affect your process and should be measured before. The actual positioning accuracy and repeatability should be discussed with regards to the geometrical accuracy of the manufactured components.
- There are some critical information missing which do not allow to replicate the results.
Results:
- How is the geometry in Figure 15 obtained?
- What is the positioning accuracy and repeatability of your machining set-up?
Discussion and conclusion:
- The discussion part is completely non-existent! This makes the research work a mere technical report and cannot be accepted for publication. A reference to other works, including a reference to machining of thin walled structures made of conventionally prepared substrate 316L stainless steel is a must!
- The paper describes the results but does not provide the in-depth analysis of the phenomena involved in this study. It should present some hypothesis of why certain effects were noted.
Reviewer 2 Report
Paper deals with the problem of additive manufacturing of thin walled parts made of 316\l steel. The aim of the work was to optimise the proces parameters to produce parts with minimum distortion and residual stress. \for the latter, hybrid process was introduced. The vast part of the manuscript is focused on computer modelling using fem. Paper is well written and structutred. It can be accepted after small corrections have been made:
1) please provide justification of what is better/new in the modeling approach that you used with respect to previously pubilshed results? Authors state in the introduction that so far the modeling of 3d manufacruring was characterized of poor acuracy but in the current work the discrepances between calculated and measured results (temperature difference) is also high. This short justification would strengthen further the already high quality of the paper.
Reviewer 3 Report
In this paper, thin-walled parts from 316L stainless steel powder are additively manufactured via Additive-Subtractive Hybrid Manufacturing (ASHM). Both thermal stress deformation of additively manufactured parts and the deformation of these parts after milling were measured. The transient temperature field and stress distribution during the ASHM of the 316L stainless steel thin-walled parts were simulated via the finite element software, ANSYS. This constructed model was also used to predict the final deformation of the thin-walled parts. The current paper is worthy of investigation and organization. The following comments need to be addressed to be accepted for publication.
- The proficiency of the language needs more improvement in the manuscript.
- The thermal and physical properties in Table 3, how many samples were used? Errors should be added. Yield strength and young’s modulus are mechanical properties.
- I recommend adding the microstructure of the as-built samples, how the processing parameters affect the microstructure. The microstructure plays a great role on the properties of the materials.
The results of FES are well presented. I recommend accepting this paper after minor revision
Round 2
Reviewer 1 Report
Thank you very much for improving your paper.
Still I cannot find the correct statistical analysis of your results as presented in Figure 6b. When looking at box-and-whisker plots I suppose the differences between top bottom and middle layer are not statistically relevant. Please conduct ANOVA analysis (with testing prior normality of residuals) and post-hoc tests to determine that 1) location is a statistically relevant factor and 2) the differences between each pair (bottom vs top/ top vs middle etc) are also statisically relevant. Please make changes in the results/discussion and conclusion according to achieved p-value.
